# LONGVTG-R1: REINFORCEMENT LEARNING FOR ROBUST LONG-VIDEO TEMPORAL GROUNDING

## ABSTRACT

We present the first Reinforcement Learning (RL)-based framework that equips Multimodal Large Language Models (MLLMs) with long-video temporal grounding skills, and demonstrate that this approach also generalizes to improve performance on general question-answering (QA) tasks. Unlike dominant supervised fine-tuning (SFT) methods, RL enables models to acquire temporal grounding abilities without risking catastrophic forgetting of their core understanding. However, adopting RL for long-video temporal grounding reveals a challenge in balancing *exploitation* of pre-trained knowledge with *exploration* of new localization skills. To address this, we propose *Token-aware KL Regularization*, which selectively relaxes the KL-divergence regularization on timestamp-related tokens to guide exploration. Moreover, effective optimization requires a learning signal that alleviates the *sparsity* of key events in long videos, for which we introduce a denser reward, the *Center Distance Reward* (CenDist). To further mitigate grounding ambiguity between language queries and visually similar content, and to facilitate effective RL training, we propose an automatic data construction method and construct a small while high-quality *SceneTG*. Our resulting model, LONGVTG-R1, delivers substantial improvements across three long-video temporal grounding datasets among efficiently fine-tuned MLLMs, and even approaches the performance of densely pre-trained or continually trained models. Beyond temporal grounding, we further verify its generalization to long-video QA: under a "Ground-then-Answer" strategy, LONGVTG-R1 consistently enhances downstream QA performance, serving as an effective first-stage grounding module.

## 1 INTRODUCTION

Understanding long videos, where models must reason over hours of untrimmed content, has become an essential goal for Multimodal Large Language Models (MLLMs). A central step in this process is temporal grounding (Gao et al., 2017; Zhang et al., 2023; Liu et al., 2025), the task of localizing fine-grained video segments that correspond to a natural language query (*e.g.*, "Find the moment when the driver releases the steering wheel"). Despite its importance, temporal grounding in long videos remains a significant bottleneck due to two factors: (1) the **sparsity** of query-relevant events in long videos indicates that even a 1% shift in the total duration can cause a drastic miss of the targets; and (2) the **ambiguity** of language queries for visual contents necessitates distinguishing visually similar and recurring events by leveraging long video context. Since temporal grounding requires highly specific timestamps, such **sparsity** and **ambiguity** challenge MLLMs to *precisely* comprehend long videos.

While the dominant paradigm for adapting MLLMs is Supervised Fine-Tuning (SFT), we contend that Reinforcement Learning (RL) offers a more principled approach for temporal grounding. With MLLMs already possessing foundational abilities, RL can incentivize a model to acquire the new, precise skill of localization without risking catastrophic forgetting of its core understanding, a known issue in SFT (Huang et al., 2024c; Mi et al., 2020; Li et al., 2025a). Crucially, temporal grounding provides naturally verifiable supervision through timestamps, making it ideal for Reinforcement Learning with Verifiable Rewards (RLVR) (Guo et al., 2025a; Gao et al., 2024), the same paradigm that has successfully encouraged deeper reasoning in advanced Large Language Models (LLMs). We therefore adopt RL as the natural optimization framework to unlock precise grounding in long

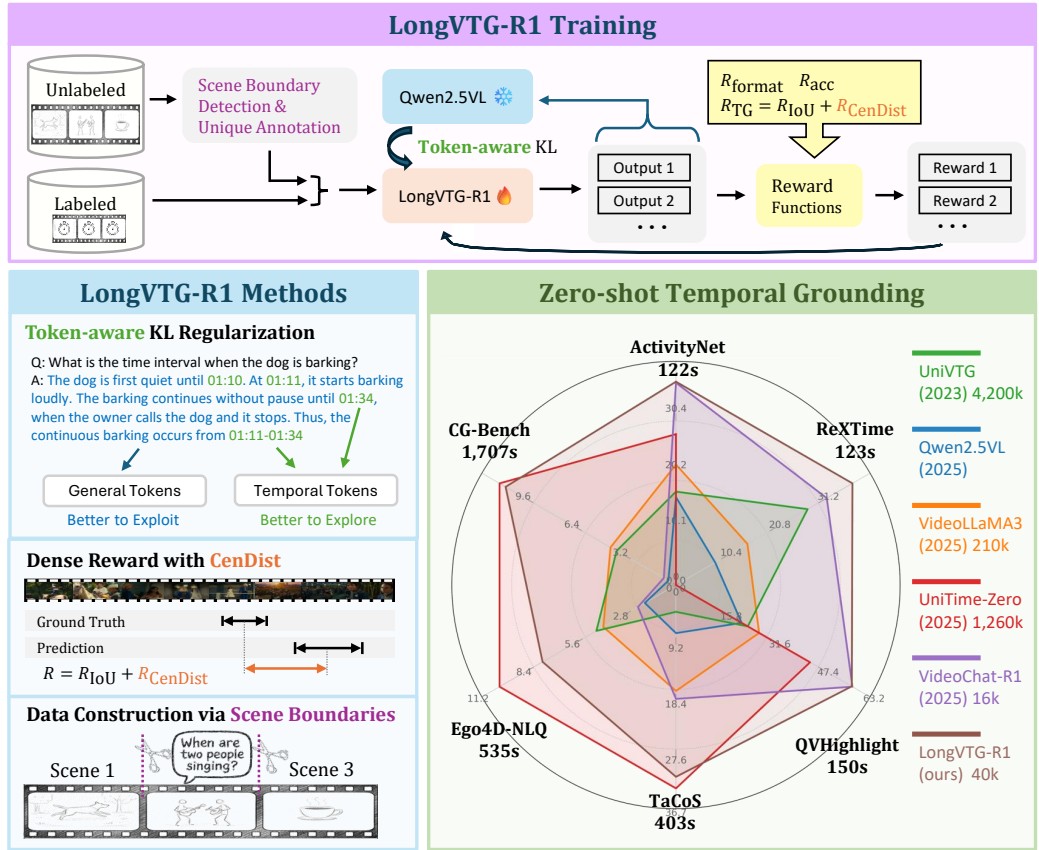

Figure 1: Overview of LONGVTG-R1. **(Top)** To enable effective RL fine-tuning for long video temporal grounding, we first automatically annotate unlabeled data via scene boundary detection and merge it with labeled data to form the training set. Then, we apply GRPO-based RL fine-tuning with our proposed token-aware KL regularization, and introduce CenDist reward to provide denser learning signals. **(Bottom-Left)** Our main technical contributions are: (i) token-aware KL regularization for balancing exploration and exploitation, (ii) dense reward design with CenDist to reduce reward sparsity, and (iii) the SceneTG dataset constructed via scene boundaries for sharp and unambiguous grounding. **(Bottom-Right)** LONGVTG-R1 is comparable to UniTime-Zero with 31x less training data, and outperforms all other models, especially on longer benchmarks.

videos, thereby strengthening the broader video understanding capabilities of MLLMs through verifiable rewards from temporal grounding.

Adopting the RL framework for long-video temporal grounding, however, reveals a fundamental trade-off between *exploration* and *exploitation*: the model must explore novel strategies for generating precise, structured timestamps to address event sparsity and ambiguity, while exploiting its vast pre-trained knowledge for semantic perception and language understanding, which forms the foundation of its reasoning. Conventional KL-divergence regularization, as used in PPO (Schulman et al., 2017) and GRPO (Shao et al., 2024), applies a uniform penalty to the learning objective across the entire vocabulary and is thus ill-suited to this setting. In temporal grounding as illustrated in Figure 1, *the exploration-critical tokens are clearly identifiable: those representing timestamps*. We therefore propose *Token-aware KL Regularization*, which resolves the exploration-exploitation tension by selectively relaxing the penalty on timestamp-related tokens (*e.g.*, digits and separators) to encourage exploration, while maintaining stronger penalties on general language tokens to preserve core perceptual capabilities.

Besides the exploration-exploitation trade-off, the inherent *sparsity* of the reward in temporal grounding further hinders MLLMs from receiving meaningful feedback during exploration toward precise grounding. This sparsity arises because the standard Intersection-over-Union (IoU) reward provides no learning gradient for predictions that are temporally close but non-

overlapping (Rezatofighi et al., 2019), which is a common scenario in long videos. To provide a denser learning signal, we introduce the *Center Distance Reward* (CenDist), an auxiliary reward that encourages predictions to approach ground truth segment centers. Beyond alleviating sparsity, we also enhance the quality of reward signals by addressing the *ambiguity* problem, particularly targeting the noisy patterns in existing datasets (Xia et al., 2023) that often contain imprecise boundaries and queries matching multiple segments. To provide high-quality training data with minimized grounding ambiguity, we automatically construct the *SceneTG* dataset through a "propose-then-annotate" paradigm, which leverages scene boundary detection and a uniqueness filter to produce temporally sharp and semantically unambiguous data for RL.

To the best of our knowledge, this work presents the *first* RL-based framework for *long-video* temporal grounding, named LONGVTG-R1 (Figure 1), which also demonstrates generalization to long-video question-answering (QA). Specifically, we show that: (1) LONGVTG-R1 substantially improves performance on three long-video temporal grounding benchmarks among efficiently fine-tuned MLLMs, approaching the performance of densely pre-trained or continually trained models. (2) It can be seamlessly integrated into existing long-video QA approaches by first localizing key segments for off-the-shelf MLLMs. (3) The temporal grounding skills acquired through RL *generalize to general long-video QA tasks* with a "Ground-then-Answer" strategy. In summary, our contributions are as follows:

- We propose token-aware KL regularization to improve the exploration-exploitation trade-off in RL. Although introduced in the context of long-video temporal grounding, our theoretical analysis suggests it is broadly applicable to RL fine-tuning methods, and we hope it can benefit other tasks in future work.

- We introduce an auxiliary CenDist reward, which provides denser learning signals and alleviates the reward sparsity challenge in long-video temporal grounding.

- We design an automatic "propose-then-annotate" paradigm for constructing temporal grounding datasets, leading to the SceneTG dataset with sharp boundaries and unambiguous queries that facilitate effective RL training.

- We demonstrate successful RL fine-tuning for long-video temporal grounding, with the resulting model, LONGVTG-R1, achieving notable gains on long-video benchmarks and improving long-video QA when used as a grounding module.

## 2 RELATED WORK

**Video Temporal Grounding.** The task of Video Temporal Grounding (VTG) aims to localize temporal segments within untrimmed videos given natural language queries. Early approaches are predominantly feature-based, relying on pretrained encoders (Radford et al., 2021; Carreira & Zisserman, 2017; Devlin et al., 2019) to extract multimodal features before applying timestamp regression modules. However, these methods are constrained by the quality of frozen features and often struggle to capture fine-grained temporal cues. More recent methods adopt end-to-end designs, treating temporal grounding as an autoregressive generation task with Multimodal Large Language Models (MLLMs). For example, frameworks such as Moment-DETR (Lei et al., 2021) reformulate grounding as a set prediction problem, while others leverage sequence generation strategies to produce event boundaries jointly with saliency and captions (Li et al., 2024; Ren et al., 2024)). Other works such as SnAG (Mu et al., 2024), Grounding-MD (Di & Xie, 2024), and LITA (Huang et al., 2024b) further push scalable or unified VTG modeling for long-form or multi-task grounding scenarios.

**Temporal Grounding with Multimodal Large Language Models (MLLMs).** Recent advances in multimodal large language models (MLLMs) have stimulated growing interest in temporal grounding as a testbed for temporal reasoning. Existing temporal MLLMs can be categorized according to how models perceive temporal information. Implicit temporal models inject time into the representation space via temporal embeddings or relative positional encodings. For instance, TimeChat (Ren et al., 2024) and TimeSuite (Zeng et al., 2024) incorporate timestamp-aware encoders, VTG-LLM employs absolute time embeddings (Guo et al., 2025b), and Qwen2.5VL (Bai et al., 2025) integrates temporal modeling through MRoPE. Explicit timestamp marking models augment the input with textualized time indicators. Methods such as Mr.BLIP (Meinardus et al., 2024), TimeMarker (Chen et al., 2024c), VideoLLaMA3 (Zhang et al., 2025), and UniTime (Li et al., 2025b) prepend tex-

tual timestamps or scene markers to video frames. Beyond architectural choices, conventionally, MLLMs achieve temporal reasoning implicitly during pretraining or continual training. Hence, another line of research focuses on efficient fine-tuning strategies. VideoChat-R1 (Li et al., 2025a) and Time-R1 (Wang et al., 2025a) inherit the implicit design of Qwen2.5VL and enhance fine-grained temporal reasoning through reinforcement-based optimization.

**Reinforcement Fine-tuning for MLLMs.** Reinforcement learning (RL) has recently emerged as a powerful paradigm for enhancing the reasoning capabilities of both LLMs and MLLMs. Classical approaches such as RLHF focus on aligning with human preferences (Christiano et al., 2017; Ouyang et al., 2022), while verifiable reward mechanisms (RLVR) exploit deterministic objectives like mathematical correctness or timestamp accuracy (Shao et al., 2024; Wen et al., 2025). In the vision-language domain, RL-based post-training has been applied to tasks including visual reasoning, spatial grounding, and short-video understanding (Meng et al., 2025; Yan et al., 2025), with methods such as TimeZero, R1-Omni, and VideoR1 showing the potential of GRPO-style optimization for temporal reasoning (Wang et al., 2025b; Rouditchenko et al., 2025; Feng et al., 2025). However, long-form video understanding poses additional challenges: the relevant segments are sparse, timestamp errors are amplified. Our work extends this line of research by adapting RLVR to long-video temporal grounding.

## 3 METHODOLOGY

### 3.1 PRELIMINARY

Group Relative Policy Optimization (GRPO) is a reinforcement post-training method proposed to enhance Large Language Models reasoning capability by Deepseek-math (Shao et al., 2024), and has recently been adapted for short temporal grounding in models like VideoChat-R1 (Li et al., 2025a) and Time-R1 (Wang et al., 2025a). In this setting, the policy ($\pi_\theta$) and reference ($\pi_{\text{ref}}$) models are MLLMs, and the query $q$ consists of a question and a video. The policy model $\pi_\theta$ generates a group of $G$ responses $o = \{o_1, \ldots, o_G\}$, through policy sampling. Each $o_i = (o_{i,1}, \ldots, o_{i,|o_i|})$ is a token sequence, where $o_{i,t}$ denotes the token generated at step $t$. A rule-based reward function measures the quality of the output, which has three components,

$$R = R_{\text{format}} + R_{\text{acc}} + R_{\text{tg}}, \tag{1}$$

where $R_{\text{format}}$ ensures valid output formatting, $R_{\text{acc}}$ measures question answering correctness, and $R_{\text{tg}}$ captures temporal grounding quality. The normalized advantage $\hat{A}_{i,t}$ is estimated from these rewards after normalization across sampled completions.

With the normalized advantages, GRPO optimizes the policy $\pi_\theta$ by maximizing the following objective function:

$$\mathcal{J}_{\text{GRPO}}(\theta) = \mathbb{E}_{\{o_i\}_{i=1}^G \sim \pi_\theta(O|q)} \left[ \frac{1}{G} \sum_{i=1}^G \frac{1}{|o_i|} \sum_{t=1}^{|o_i|} \left\{ \min\left(r_{i,t}(\theta)\,\hat{A}_{i,t},\; \text{clip}(r_{i,t}(\theta),\, 1-\varepsilon,\, 1+\varepsilon)\,\hat{A}_{i,t}\right) \right. \right.$$
$$\left. \left. -\; \beta\, D_{\text{KL}}\big(\pi_\theta(\cdot \mid q, o_{i,<t}) \,\|\, \pi_{\text{ref}}(\cdot \mid q, o_{i,<t})\big) \right\} \right], \tag{2}$$

where $r_{i,t}(\theta)$ denotes the probability ratio between the policy model and the reference model, and $D_{\text{KL}}$ is the regularization term measuring the distance between the two distributions:

$$D_{\text{KL}}\big(\pi_\theta \,\|\, \pi_{\text{ref}}\big) = \sum_{o_{i,t}} \pi_\theta(o_{i,t}) \log \frac{\pi_\theta(o_{i,t})}{\pi_{\text{ref}}(o_{i,t})}. \tag{3}$$

### 3.2 TOKEN-AWARE KL REGULARIZATION

The standard KL regularization term in the GRPO objective (Equation 3) measures the distance between the two distributions, serves as a penalty to prevent the policy $\pi_\theta$ from deviating catastrophically from the reference model. However, its uniform penalty across the entire vocabulary (i.e., token-agnostic) creates a dilemma that directly corresponds to the exploration-exploitation trade-off in temporal grounding, especially when we can clearly separate the tokens encouraged for exploration, namely those representing timestamps. This approach fails to distinguish between tokens

that require exploration (e.g., digits and separators for generating novel, precise timestamps) and those that benefit from exploitation (i.e., leveraging the vast linguistic knowledge of the pre-trained MLLM). Consequently, the uniform KL regularization serving as a penalty term in the learning objective over-penalizes experimentation with timestamp tokens, hindering the model's ability to learn precise temporal structures.

A naive approach is to mask out timestamp tokens in the KL sum. However, this is mathematically unsound. Specifically, calculating the KL divergence over a selected subset $S$

$$\sum_{o \in S} \pi_\theta(o) \log \frac{\pi_\theta(o)}{\pi_{\mathrm{ref}}(o)}$$

leads to an unexpected optimal point. It reaches a negative minimum value $-\pi_{\mathrm{ref}}(S)/e$ when $\pi_\theta(o) = \pi_{\mathrm{ref}}(o)/e$ for all $o \in S$. [1]

To correct the optimization, we introduce *token-aware KL*, which imposes a principled, subset-aware divergence that preserves *non-negativity* and *identity of indiscernibles* on the subset.

**Definition.** For any non-empty $S$ in any discrete space and distributions $p, q$ on the same domain, define

$$D_{\mathrm{KL}}^S(p \,\|\, q) \;\triangleq\; q(S) - p(S) + \sum_{o \in S} \Big( p(o) \log \frac{p(o)}{q(o)} \Big). \tag{4}$$

This is the form of the KL restricted to $S$, augmented with a linear correction term, $q(S) - p(S)$, which ensures proper behavior as a penalty of $p(S) \neq q(S)$.

**Basic properties.** For any $S \neq \varnothing$ and any discrete $p, q$:

1. (*Non-negativity*) $D_{\mathrm{KL}}^S(p \,\|\, q) \geq 0$.
2. (*Identity of indiscernibles on $S$*) $D_{\mathrm{KL}}^S(p \,\|\, q) = 0$ iff $p(o) = q(o)$ for all $o \in S$.

The proof of the properties is given in Appendix B. This theoretical analysis suggests that token-aware KL regularization is broadly applicable to RL methods when the action space is discrete.

**Token-aware KL to Temporal Grounding** We partition the vocabulary into timestamp tokens versus everything else:

$$S_{\mathrm{time}} \;=\; \{\text{digits, time separators, timestamp tokens}\}, \qquad S_{\mathrm{text}} \;=\; \mathcal{V} \setminus S_{\mathrm{time}}.$$

Fine-grain KL replaces the uniform KL in GRPO with a weighted sum over these subsets:

$$\beta_{\mathrm{text}} \, D_{\mathrm{KL}}^{S_{\mathrm{text}}}\Big( \pi_\theta(\cdot \mid q, o_{<t}) \,\big\|\, \pi_{\mathrm{ref}}(\cdot \mid q, o_{<t}) \Big) \;+\; \beta_{\mathrm{time}} \, D_{\mathrm{KL}}^{S_{\mathrm{time}}}\Big( \pi_\theta(\cdot \mid q, o_{<t}) \,\big\|\, \pi_{\mathrm{ref}}(\cdot \mid q, o_{<t}) \Big), \tag{5}$$

We set a strong regularization coefficient $\beta_{\mathrm{text}} = 0.04$ to exploit the reference model's robust language capabilities, ensuring fluency and semantic coherence. Conversely, we set $\beta_{\mathrm{time}} = 0$ to completely relax the penalty on timestamp tokens, thereby encouraging the model to explore and learn novel, precise temporal grounding structures without being constrained by the potentially flawed priors of the reference model.

### 3.3 Dense Reward Design with CenDist

Besides the KL-divergence regularization, the inherent sparsity of the long video temporal grounding task prevents the model from receiving dense reward signals and effectively learn from guidance. Specifically, while IoU is widely used as the standard evaluation metric (Lei et al., 2021; Regneri et al., 2013), it provides a zero learning gradient for predictions that are temporally close but do not overlap with the ground truth, often in long videos and in the early training stage.

To provide a denser reward signal, we introduce the Center Distance reward ($R_{\mathrm{CenDist}}$), which measures the distance between the centers of the predicted and ground-truth intervals. Let $[l_{\mathrm{gt}}, r_{\mathrm{gt}}]$

---

[1]Each term $p(o) \log \frac{p(o)}{q(o)}$ is minimized at $p(o) = q(o)/e$ with value $-q(o)/e$, the sum attains $-q(S)/e$.

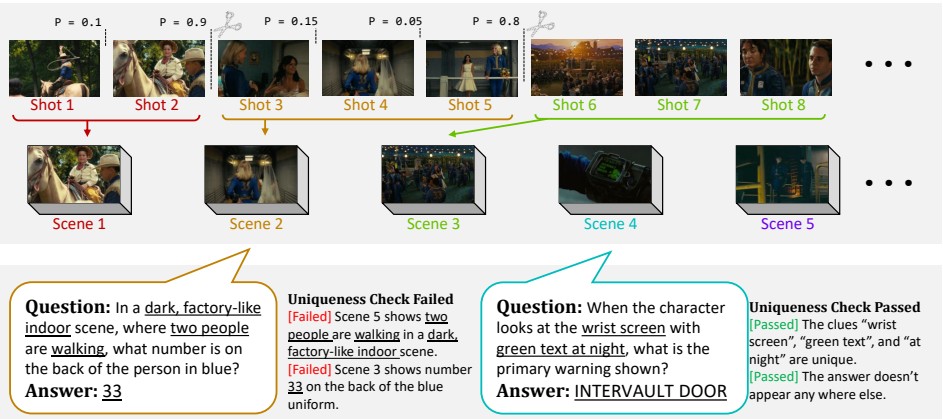

Figure 2: **Overview of SceneTG data construction.** We begin with unlabeled videos and first apply ShotCoL to detect scene boundaries. Next, we generate question-answer pairs for each identified scene. Finally, we perform a uniqueness check using two criteria to filter out queries that are relevant to multiple scenes.

and $[l, r]$ be the ground-truth and predicted intervals, respectively:

$$R_{\text{CenDist}} = 1 - \frac{\left| \frac{l_{\text{gt}} + r_{\text{gt}}}{2} - \frac{l+r}{2} \right|}{\text{Duration}}. \tag{6}$$

The total temporal grounding reward is then:

$$R_{\text{tg}} = \alpha \cdot R_{\text{IoU}} + (1 - \alpha) \cdot R_{\text{CenDist}}. \tag{7}$$

### 3.4 SceneTG: Clear and Unambiguous Temporal Grounding Dataset

Finally, we investigate how to provide high-quality reward signals and enable the model to mitigate the *ambiguity* challenge in long video grounding. Such a challenge mainly arises from the *ambiguous* patterns in existing datasets, reflected from two perspectives: (i) Imprecise boundaries. Conventional temporal grounding datasets follow a "caption-then-localize" paradigm, requiring substantial human effort to align captions with video spans. However, human annotations rarely achieve millisecond- or frame-level precision. (ii) Queries matching multiple segments. Long-form videos often contain recurring events or settings, a single query may correspond to multiple segments.

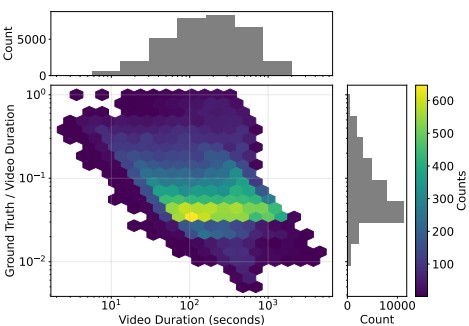

Figure 3: **SceneTG statistics** of video durations with ground truth density

To address this challenge, we propose an automatic data construction method that can be applied to any collection of long-form videos. We then build the SceneTG dataset with our internal data of 20K unlabeled video segments from movies and TV contents. To ensure **(i) Boundary Clarity** and **(ii) Content Uniqueness**, we adopt a "propose-then-annotate" paradigm grounded in scene boundaries, and apply a uniqueness check to filter out queries with recurrent patterns, as illustrated in Figure 2.

**Scene Boundaries as Candidates.** To ensure boundary clarity, we first leverage natural scene breaks within videos. Our "propose-then-annotate" paradigm reverses the typical workflow: we first identify candidate segments based on scene cuts and then generate descriptive queries for them. Scenes are the natural temporal units of long-form video, grouping multiple shots into a coherent block by location, characters, or intent. Anchoring on scenes guarantees that neighboring segments do not drift into one another, thereby enhancing boundary clarity. In practice, we reproduce ShotCoL (Chen et al., 2021) to detect scene boundaries by classifying shot transitions. Formally, a video $V$ is decomposed into shots $\{s_i\}_{i=1}^{N}$, and the ShotCoL detector assigns each boundary $b_i$ a probability $p_i$ of

being a scene cut. We threshold $p_i > 0.3$ and merge shots between consecutive scene cuts to form candidate segments.

**Uniqueness Check Filters out Recurrent Clues.** For each proposed scene segment, we use Claude-3.7 to generate question–answer (QA) pairs specifically prompted to elicit discriminative details—such as unique objects, fine-grained actions, or context-specific interactions. Each QA pair is then subjected to a uniqueness check with two conditions: none of the surrounding ten scenes (five before and five after) should (i) contain all the clues in the question, nor (ii) contain the answer. QA pairs that fail this test are discarded, ensuring that no query admits multiple valid matches even across distant parts of the video.

Figure 3 summarizes the key statistics of the SceneTG dataset. Most videos range from 100 to 1,000 seconds in length, and the annotated ground-truth segments constitute 3%–10% of the total video length.

## 4 EXPERIMENTS

### 4.1 IMPLEMENTATION DETAIL

We build upon Qwen2.5VL-7B as the base model, fine-tuning on a mixture of three temporal grounding datasets, Charades-STA (Gao et al., 2017) and NExT-GQA (Xiao et al., 2024), and our SceneTG. We set $\alpha = 0.8$ when mixing $R_{\text{CenDist}}$ and $R_{\text{IoU}}$. The training time is 90 hours with 8 NVIDIA B200 GPUs.

### 4.2 TEMPORAL GROUNDING EVALUATION

We conduct both **in-domain** and **out-of-domain (zero-shot)** evaluations on temporal grounding benchmarks. The metric is the mean Intersection-over-Union (mIoU) between predicted and ground-truth spans.

Table 1 shows the evaluation results on in-domain benchmarks, Charades (Sigurdsson et al., 2016) and

Table 1: In-domain temporal grounding evaluation.

| Model | In-domain Evaluation | |
|---|---|---|
| | Charades | NextGQA |
| Video-XL2-8B (Qin et al., 2025) | 54.2 | – |
| Videollama3 (Zhang et al., 2025) | 59.9 | – |
| Time-R1 (Wang et al., 2025a) | 58.1 | – |
| VideoChat-R1 (Li et al., 2025a) | 60.8 | 32.4 |
| LONGVTG-R1 | **61.5** | **38.2** |
| VideoChat-R1 (Fair Comparison) | 59.8 | 35.6 |

NextGQA (Xiao et al., 2024). LONGVTG-R1 achieves the best results among compared models, reaching 61.5 on Charades and 38.2 on NextGQA, surpassing Videollama3 and VideoChat-R1, demonstrating the effectiveness of our training methods.

Beyond in-domain performance, we further evaluate on six out-domain benchmarks: ActivityNet (Caba Heilbron et al., 2015), ReXTime (Chen et al., 2024b), QVHighlight (Lei et al., 2021), TaCoS (Regneri et al., 2013), Ego4D-NLQ (Grauman et al., 2022), and CGBench (Chen et al., 2024a), which cover both short- and long-video scenarios. All evaluations are conducted under a zero-shot setting—none of the models, including LONGVTG-R1, are fine-tuned on these benchmarks. As shown in Table 2, LONGVTG-R1 performs on par with the recent UniTime-Zero model while using 31× less data (40k vs. 1,260k). Compared with all other models, LONGVTG-R1 achieves substantial gains on long-video temporal grounding tasks, highlighting the effectiveness of our design for long videos. A full comparison table—including (1) additional metrics (Rec@0.3/0.5/0.7), (2) more baselines (*e.g.*, Video-XL2 (Qin et al., 2025)), and (3) in-domain models (*e.g.*, SnAG (Mu et al., 2024))—is provided in Appendix A.

### 4.3 GENERAL LONG VIDEO QA EVALUATION

Beyond temporal grounding, we examine whether the reinforcement fine-tuning on grounding also brings benefits to general long-video question answering (QA). We evaluate on four representative benchmarks: LongVideoBench (Wu et al., 2024), MLVU (Zhou et al., 2025), VideoMME (Long) (Fu et al., 2025), and LVBench (Wang et al., 2024).

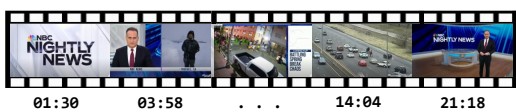

**01:30**     **03:58**    **. . .**     **14:04**     **21:18**

**Question:** In a screen with a blue border, there is a highway. On the highway, many cars are parked. Two horses of different colors appear on the highway. What did the yellow-brown horse do the first time it appeared?
(A)... (B) Ran against the direction of the traffic flow. (C)...
(D)... (E) Ran in the direction of the traffic flow

**Input Full Video**

\<think\>
The video clip starts with a news segment about a dangerous blizzard, then transitions to a highway scene where two horses are seen on the road. The horses are not interacting with any vehicles or crossing the highway at this point. Therefore, none of the given options describe the actions of the yellow-brown horse the first time it appeared.
\</think\>
\<answer\>**E (Wrong)**\</answer\>

**Temporal Grounding**

\<timestamp\>834.16 to 862.59 seconds\</timestamp\>
**Input Video Clip (13:54-14:23)**

\<think\>The video starts with a news anchor reporting about thieves using new technology to steal cars. The scene then shifts to a crowded street at night where people are running away from a pickup truck. Next, an elderly woman wearing a yellow outfit is shown. The video then cuts to a news anchor reporting about horses on the loose stopping highway traffic. The horses are seen running across the highway, causing traffic to stop.\</think\>
\<answer\>**B (Correct)**\</answer\>

Figure 4: **Qualitative Analysis.** The input video lasts over 20 minutes and includes various news segments. The question requires identifying the first action of the yellow-brown horse when it appeared on a highway scene. Directly answering from the full video (left) leads to a wrong prediction since the model fails to capture the short but critical moment, concluding that the horse did not interact with vehicles. By contrast, temporal grounding (right) successfully localizes the correct short clip (13:54–14:23), where the horses are running across the highway, yielding the correct answer. This example illustrates how long video temporal grounding enhances long-video question answering by focusing attention on the most relevant segment.

**RL preserve, SFT forget.** We first study whether LONGVTG-R1 can directly answer QA questions when given full videos. As shown in Table 3, compared with Qwen2.5VL, our LONGVTG-R1 achieves modest improvements on the three longer benchmarks (MLVU, VideoMME, and LVBench). However, on the shortest benchmark, LongVideBench, the effect appears inconsistent. This similar pattern is also observed on VideoChat-R1. This indicates RL preserves most QA capability. On the contrary, SFT results in a model that cannot answer the question in a correct format.

**Grounding can be an intermediate step of QA.** We next investigate whether temporal grounding can serve as an explicit intermediate step to improve QA. In this setting, a grounding model first localizes the relevant segment, and Qwen2.5VL then answers questions based only on this grounded clip. As shown in Table 3, only LONGVTG-R1 consistently boosts QA accuracy compared to using the entire video, while UniTime-Zero brings less improvement, Qwen2.5VL and VideoChat-R1 even generally provide negative effect compared with the directly answering way.

**Unifying grounding and QA in a single model.** Finally, we test whether a single model can handle both temporal grounding and QA jointly. In this unified method, LONGVTG-R1 is used for both localization and question answering. Table 3 shows that this unified method achieves performance

Table 2: Zero-shot Temporal Grounding on long videos and short videos. Note that UniTime-Zero is just for reference, not for comparison, as it uses 31x more data.

| Model | Temporal Training Data | Long Videos | | | Short Videos | | |
|---|---|---|---|---|---|---|---|
| | | TaCoS 403s | Ego4D 535s | CGBench 1707s | ANet 122s | ReXTime 123s | QVHigh 150s |
| Qwen2.5VL (Bai et al., 2025) | | 7.9 | 1.8 | 0.5 | 15.9 | 8.4 | 21.4 |
| Videollama3 (Zhang et al., 2025) | 210k | 17.4 | 4.2 | 4.3 | 21.8 | 15.3 | 27.1 |
| UniVTG (Lin et al., 2023) | | 4.4 | 4.6 | 3.9 | 16.9 | 28.2 | 23.4 |
| Mr. BLIP (Meinardus et al., 2024) | | 17.9 | 5.4 | | | | |
| VTG-LLM (Guo et al., 2025b) | | 5.3 | 1.4 | | 17.9 | | 8.6 |
| VTimeLLM (Huang et al., 2024a) | | 8.1 | 2.5 | 1.6 | 29.3 | 20.1 | |
| Timechat (Ren et al., 2024) | | 3.0 | 1.3 | | 21.5 | 11.7 | 14.3 |
| TimeMaker (Chen et al., 2024c) | 222k | | | | 22.0 | | 21.3 |
| TimeSuite (Zeng et al., 2024) | 349k | 5.7 | 0.9 | 2.1 | | | |
| VideoChat-R1 (Li et al., 2025a) | 16k | 18.7 | 2.2 | 0.8 | 36.6 | 32.3 | **57.5** |
| LONGVTG-R1 | 40k | **31.8** | **7.7** | **11.2** | **36.8** | **37.8** | 57.3 |
| UniTime-Zero (Fair Comparison) | 40k | 1.6 | 0.11 | | | 6.9 | 12.1 |
| VideoChat-R1 (Fair Comparison) | 40k | 22.5 | 4.5 | 9.6 | | 36.4 | 57.3 |
| UniTime-Zero (Li et al., 2025b) | 1,260k | 33.4 | 10.2 | 11.6 | 27.3 | | 43.7 |

Table 3: **Evaluation on Long Video QA Benchmarks** in three settings, direct, ground-then-answer, and unified. (1) In the direct setting, reinforcement-trained LONGVTG-R1 preserves general QA ability on long-video benchmarks, whereas SFT substantially degrades the model and prevents it from producing valid answers. (2) With grounding as an explicit intermediate step, LONGVTG-R1 yield consistent QA improvements. (3) In the unified setting, a single LONGVTG-R1 model can jointly perform grounding and QA without loss of performance.

| Setting | Ground Model | QA Model | LongVideoBench 473s | MLVU 651s | VideoMME (Long) 2,386s | LVBench 4,101s |
|---|---|---|---|---|---|---|
| Direct | – | Qwen2.5VL | 60.5 | 68.6 | 53.4 | 41.3 |
| | – | VideoChat-R1 | 59.2 | 69.6 | 54.3 | 40.4 |
| | – | LONGVTG-R1 | 58.7 | 69.1 | 55.0 | 42.2 |
| | – | Qwen + SFT | N/A | N/A | N/A | N/A |
| Ground-then-answer | Qwen2.5VL | Qwen2.5VL | 55.5 | 55.8 | 48.4 | 33.3 |
| | VideoChat-R1 | | 57.8 | 65.6 | 52.3 | 42.6 |
| | UniTime-Zero | | 61.2 | 70.7 | 50.4 | 42.9 |
| | LONGVTG-R1 | | 63.1 | **70.8** | 54.2 | 47.1 |
| Unified | LONGVTG-R1 | LONGVTG-R1 | **63.3** | 70.6 | **54.9** | **47.5** |

Table 4: **Ablation studies on Token-aware KL and distance rewards.** Experiments here are conducted at a smaller scale to reduce computational cost. Note that the performance differences are expected to be more pronounced at full scale, as our Token-aware KL and CenDist jointly yield substantial gains in the main results. **Left:** comparison among conventional KL, no KL, and our Token-aware KL. **Right:** comparison among IoU, GIoU, and our CenDist.

| $\beta_{\text{text}}$ | $\beta_{\text{time}}$ | NextGQA mIoU | NextGQA R@0.3 | CGBench mIoU | CGBench R@0.3 | Reward | NextGQA mIoU | NextGQA R@0.3 | CGBench mIoU | CGBench R@0.3 |
|---|---|---|---|---|---|---|---|---|---|---|
| 0.04 | 0.04 | 34.6 | 52.9 | 7.77 | 9.27 | IoU | 34.8 | 54.4 | 7.82 | 9.84 |
| 0 | 0 | 34.7 | 54.0 | 7.55 | 9.50 | GIoU | 34.9 | 53.5 | 8.16 | 9.97 |
| 0.04 | 0 | 34.8 | 54.4 | 7.82 | 9.84 | IoU + CenDist | 35.5 | 55.1 | 8.17 | 10.00 |

comparable to, or slightly better than, the two-model setting. This indicates that LONGVTG-R1 has developed not only strong grounding ability, but also preserved general QA skills, enabling a streamlined one-model solution. Figure 4 shows an example from LongVideoBench, illustrating the benefits of the unifying grounding and QA setting.

### 4.4 ABLATION STUDY

We conduct ablation studies on NextGQA (in-domain) and CGBench (out-of-domain) using a smaller variant, LONGVTG-R1-tiny, trained with only 30% data and one third of the total pixels.

**Token-aware KL.** As shown in Table 4, we compare token-aware KL with two normal variants: KL with $\beta = 0.04$ and training without any KL divergence. The results indicate that applying KL divergence to time tokens hinders model learning in the domain, in other words, the in-domain performance improves when $\beta_{\text{time}} = 0$. However, the benefits generalize to out-of-domain benchmarks only when we retain KL divergence on semantic tokens, that is, when $\beta_{\text{text}} \neq 0$.

**CenDist.** Table 4 shows the comparison among IoU, GIoU, and our CenDist. The results demonstrate that CenDist—where the distance is normalized by duration—should be considered the preferred option for training long-video temporal grounding models.

## 5 CONCLUSION

In this work, we introduced the reinforcement learning framework for enhancing multimodal LLMs in long-video temporal grounding. Our approach integrates token-aware KL regularization for balancing exploration and exploitation, center distance reward for alleviating reward sparsity, and the constructing dataset SceneTG with sharp boundaries and unambiguous queries. The resulting model, LONGVTG-R1, achieves decent performance across multiple temporal grounding benchmarks and shows benefits to long-video QA.

## ETHICS STATEMENT

This work focuses on advancing algorithms for long video temporal grounding, with the primary goal of improving MLLMs' performance and robustness.

As part of this work, we constructed the SceneTG dataset. We are actively evaluating the dataset to ensure that no videos contain personally identifiable or sensitive information. The dataset will only be shared in compliance with ethical standards and legal requirements.

In line with the ICLR Code of Ethics, we are committed to responsible stewardship of research, transparency in methodology, and minimizing potential risks associated with dataset usage.

## REPRODUCIBILITY STATEMENT

We include main training code in supplementary materials, and include the evaluation prompts in Appendix C.

## THE USE OF LARGE LANGUAGE MODELS (LLMS)

We used LLMs to help polish the writing. We maintain records of the chat history with the LLMs, and we verified all text generated by the LLMs. We are responsible for all text in this paper.

We did not use LLMs for any part of the paper beyond writing. The core research, experimental design, and all scientific claims remain our original work. The LLMs' contributions were limited to improving the clarity of the text.

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

## A    ADDITIONAL EXPERIMENT RESULTS

| | ActivityNet (val) / tacos | | | | rextime / ego4d-nlq (val) | | | | QVHighlight (val) / CGBench | | | |
|---|---|---|---|---|---|---|---|---|---|---|---|---|
| | mIoU | R@0.3 | R@0.5 | R@0.7 | mIoU | R@0.3 | R@0.5 | R@0.7 | mIoU | R@0.3 | R@0.5 | R@0.7 |
| Qwen2.5VL | 15.9 | 20.9 | 12.1 | 6.5 | 8.36 | 10.8 | 6.84 | 4.67 | 21.4 | 27.6 | 17.0 | 9.23 |
| | 7.92 | 11.8 | 5.97 | 1.67 | 1.83 | 1.83 | 1.01 | 0.59 | 0.52 | 0.57 | 0.33 | 0.23 |
| Videollama3 | 21.8 | 28.2 | 15.6 | 7.5 | 15.3 | 19.3 | 9.23 | 4.8 | 27.1 | 29.7 | 17.9 | 11.6 |
| | 17.4 | 24.8 | 14.2 | 5.6 | 4.24 | 6.27 | 2.84 | 1.26 | 4.25 | 5.43 | 2.8 | 1.03 |
| UniVTG | 16.86 | 11.1 | 4.06 | | 28.17 | 41.34 | 26.88 | | 23.38 | 14.13 | 4.42 | |
| | 4.4 | 5.17 | 1.27 | | 4.63 | 6.48 | 3.48 | | 3.86 | | | |
| Mr. BLIP | 17.94 | 24.59 | 14.32 | | 5.37 | 6.49 | 3.2 | | | | | |
| VTG-LLM | 17.86 | 12.32 | 6.74 | | | | | | 8.6 | 4.13 | 1.55 | |
| | 5.27 | 6.87 | 2.92 | | 1.36 | 1.71 | 0.46 | | | | | |
| VTimeLLM | 29.3 | | | | 20.14 | 28.84 | 17.41 | | | | | |
| | 8.08 | 9.3 | 3.9 | | 2.52 | 1.67 | 0.77 | | 1.58 | | | |
| Timechat | 21.53 | 16.38 | 8.36 | | 11.65 | 14.42 | 7.61 | | 14.25 | 8.32 | 4.26 | |
| | 2.95 | 3.77 | 1.6 | | 1.3 | 1.67 | 0.79 | | | | | |
| TimeMaker | 22.03 | 16.56 | 9.28 | | | | | | 21.3 | 12.32 | 9.16 | |
| TimeSuite | 5.71 | 6.75 | 2.5 | | 0.94 | 0.88 | 0.43 | | 2.09 | | | |
| UniTime-Zero | 27.31 | 22.77 | 14.14 | | | | | | 43.71 | 41.03 | 31.48 | |
| | 33.38 | 50.06 | 31.54 | | 10.18 | 14.67 | 7.38 | | 11.63 | | | |
| VideoChat-R1 | 36.6 | 33.4 | | 17.7 | 32.3 | 42.2 | 32.7 | 23.3 | 57.5 | 78.0 | 62.3 | 44.3 |
| | 18.7 | 28.6 | 16.2 | 6.17 | 2.18 | 2.68 | 1.29 | 0.57 | 0.82 | 1.03 | 0.47 | 0.2 |
| LONGVTG-R1 | 36.8 | 53.2 | 34.0 | 18.0 | 37.8 | 50.0 | 36.1 | 25.8 | 57.3 | 75.7 | 59.3 | 44.5 |
| | 31.8 | 46.4 | 31.6 | 16.1 | 7.71 | 10.0 | 6.22 | 2.99 | 11.15 | 13.97 | 7.23 | 3.03 |

## B    PROOF OF THE PROPERTIES OF TOKEN-AWARE KL

**Definition.** For any non-empty subset $S$ of the token space and discrete distributions $p, q$ on the same domain, define the *Token-aware KL*

$$D_{\mathrm{KL}}^{S}(p \,\|\, q) \triangleq q(S) - p(S) + \sum_{o \in S}\left(p(o)\log\frac{p(o)}{q(o)}\right). \tag{8}$$

**Basic properties.**

**Proposition 1.** *For any non-empty $S$ and any discrete $p, q$:*

1. *(Non-negativity)* $D_{\mathrm{KL}}^{S}(p \,\|\, q) \geq 0$.

2. *(Identity on $S$)* $D_{\mathrm{KL}}^{S}(p \,\|\, q) = 0$ *if and only if* $p(o) = q(o)$ *for all* $o \in S$.

*Proof.* Fix $q$ and write $m \triangleq p(S) \in [0, 1]$ and $q_S \triangleq q(S) \in [0, 1]$. On $S$, factor the *shape* of $p$ and $q$ from their total mass:

$$r(o) \triangleq \frac{p(o)}{m}, \quad o \in S, \qquad s(o) \triangleq \frac{q(o)}{q_S}, \quad o \in S,$$

where $r$ and (when $q_S > 0$) $s$ are distributions on $S$. We first consider the case $q_S > 0$ and then treat the boundary case $q_S = 0$.

**Case 1:** $q_S > 0$. Using $p(o) = m\,r(o)$ and $q(o) = q_S\,s(o)$ for $o \in S$,

$$\sum_{o \in S} p(o)\log\frac{p(o)}{q(o)} = \sum_{o \in S} m\,r(o)\Big(\log m + \log r(o) - \log q_S - \log s(o)\Big)$$

$$= m\log\frac{m}{q_S} + m\sum_{o \in S} r(o)\log\frac{r(o)}{s(o)}$$

$$= m\log\frac{m}{q_S} + m\,\mathrm{KL}(r\|s).$$

Therefore

$$D_{\mathrm{KL}}^S(p\|q) = \underbrace{\left(q_S - m + m\log\frac{m}{q_S}\right)}_{\phi_{q_S}(m)} + m\,\mathrm{KL}(r\|s). \tag{9}$$

For fixed $m$, the second term is minimized uniquely at $r = s$ with value $0$. Hence

$$\inf_{p|_S:\, p(S)=m} D_{\mathrm{KL}}^S(p\|q) \;=\; \phi_{q_S}(m) = q_S - m + m\log\frac{m}{q_S}.$$

Now minimize $\phi_{q_S}(m)$ over $m \in [0,1]$. A direct calculation shows

$$\phi_{q_S}'(m) = \log\frac{m}{q_S}, \qquad \phi_{q_S}''(m) = \frac{1}{m} > 0 \;\;(m>0),$$

so $\phi_{q_S}$ is strictly convex on $(0,1]$ with the unique minimizer at $m = q_S$, where $\phi_{q_S}(q_S) = 0$. Combining the two steps, the unique global minimizer of $D_{\mathrm{KL}}^S(p\|q)$ is attained when

$$m^\star = q_S \quad\text{and}\quad r^\star = s \iff p(o) = q(o) \;\forall\, o \in S,$$

and the minimum value is $0$. Since equation 9 is a sum of a nonnegative divergence and a strictly convex nonnegative function of $m$, it follows that $D_{\mathrm{KL}}^S(p\|q) \geq 0$, with equality if and only if $p|_S = q|_S$.

**Case 2:** $q_S = 0$. Then $q(o) = 0$ for all $o \in S$. If $p(o) > 0$ for some $o \in S$, the logarithmic term is $+\infty$ and hence $D_{\mathrm{KL}}^S(p\|q) = +\infty > 0$. If instead $p(o) = 0$ for all $o \in S$ (equivalently $m = 0$), every term in equation 4 is $0$, so $D_{\mathrm{KL}}^S(p\|q) = 0$. Thus non-negativity holds, and equality occurs iff $p|_S = q|_S$ (both identically zero on $S$).

The two cases together prove both claims. $\qquad\square$

## C   EVALUATION PROMPTS

**Evaluation Prompt for Charades-STA, ActivityNet, QVHighlight, and TaCoS.**

```
To accurately pinpoint the event "[EVENT]" in the video, determine the
↪   precise time period of the event.

Output your thought process within the <think> </think> tags.

Then, provide the start and end times (in seconds, precise to one decimal
↪   places) in the format "start time to end time" within the <timestamp>
↪   </timestamp> tags.
```

**Evaluation Prompt for NextGQA, ReXTime, CG-Bench, and Ego4D-NLQ.**

```
Locate the video clip that is relevant to the question:
↪   "[QUESTION][OPTIONS (if applicable)]"

Output your thought process within the <think> </think> tags and a
↪   capital letter indicating the best option within the <answer>
↪   </answer> tags.

Then, provide the timestamp of video clip that support your answer in the
↪   format "start time to end time" (in seconds, precise to one decimal
↪   places) within the <timestamp> </timestamp> tags.
```

## D   ADDITIONAL QUALITATIVE ANALYSIS

Figure 5 presents a representative example from ActivityNet that illustrates the comparative behavior of our LONGVTG-R1 and the baseline VideoChat-R1 on the temporal grounding task. This case highlights the intrinsic difficulty of grounding textual descriptions in videos containing repeated or visually similar events, which can easily mislead models toward incorrect temporal spans. While

Question: Please locate the video clip "We see the other man lift **the first man** and slam him on the sidewalk"

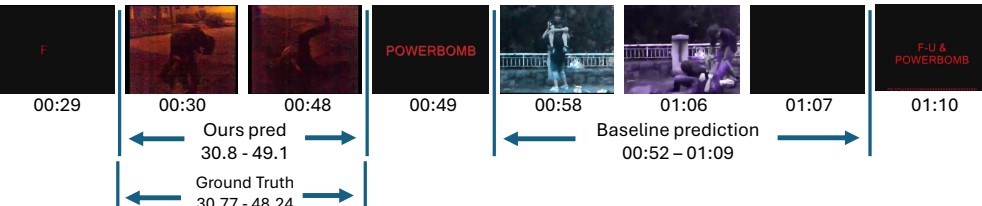

Figure 5: **Temporal Grounding Qualitative Analysis.** Comparison between LONGVTG-R1 and VideoChat-R1 on an ActivityNet example, highlighting (1) the difficulty posed by duplicate events, (2) the baseline's drift beyond scene boundaries versus our precise alignment, and (3) our model's preference for the first event when resolving "first vs. last" ambiguity.

both methods must reason over duplicate actions, the baseline exhibits temporal drift across scene boundaries, extending its predicted segment into regions where the screen has already turned black and end-screen text appears—clear evidence of misalignment with visual structure. In contrast, our model maintains precise boundaries, terminating exactly before the cut and showing stronger sensitivity to scene transitions. Moreover, when the query involves the semantic distinction between the "first" and "last" occurrence of an event—an ambiguity frequently encountered in long-form videos—LONGVTG-R1 prefers to ground the correct (first) instance, aligned with human annotations, whereas the baseline tends to select a later repetition. Together, these observations confirm our model's improved robustness to repeated events, scene-cut boundaries, and fine-grained temporal references.

