# OpenReview forum: "LongVTG-R1: Reinforcement Learning for Robust Long-Video Temporal Grounding"
_ICLR.cc/2026/Conference — Submitted to ICLR 2026_

### Official Review · Reviewer_oC5c · 2025-10-27

**Soundness:** 3
**Presentation:** 3
**Contribution:** 2
**Rating:** 4
**Confidence:** 4

**Summary:**

This paper presents LongVTG-R1, a framework applying Reinforcement Learning to enhance the long-video temporal grounding capabilities of Multimodal Large Language Models. The authors identify that standard Supervised Fine-Tuning can lead to catastrophic forgetting  and propose RL as a better alternative. The paper introduces three main technical contributions to make RL viable for this task: (1) Token-aware KL Regularization to balance exploration (for timestamps) and exploitation (for general language); (2) a Center Distance Reward to create a denser reward signal and alleviate the sparsity of the standard IoU metric ; and (3) a new dataset, SceneTG, constructed automatically using scene boundaries and a uniqueness filter to provide clear and unambiguous training signals. The authors show that their model achieves state-of-the-art results on several zero-shot VTG benchmarks and claim this skill also generalizes to improve general-purpose long-video Question-Answering.

**Strengths:**

1. Relevant Problem and Sound Motivation: The paper tackles the challenging and important problem of long-video temporal grounding. The motivation to use RL as an alternative to SFT to prevent catastrophic forgetting is sound and follows a logical trend from recent works on shorter video grounding.

2. Clear Identification of Challenges: The authors correctly identify the key bottlenecks in applying RL to this domain: the exploration-exploitation trade-off , reward sparsity , and data ambiguity. The paper is well-structured around proposing a specific solution for each of these problems.

3. Intuitive Technical Solutions: The proposed contributions are clever and intuitive. The Token-aware KL regularization, in particular, is a task-specific and sensible way to manage the KL penalty by partitioning the vocabulary. The "propose-then-annotate" paradigm for the SceneTG dataset is also a practical approach to generating higher-quality data.

4. Strong Empirical Grounding Results: The paper demonstrates strong performance on its primary task: temporal grounding. As shown in Table 2, LongVTG-R1 achieves state-of-the-art (SOTA) results across a suite of six zero-shot benchmarks. Achieving this with a smaller dataset (40k) than some competitors highlights the data efficiency of the approach.

**Weaknesses:**

1. Overstated Generalization to QA: A major weakness is the overstatement of the paper's "most surprising" finding—generalization to direct long-video QA. This claim is not well-supported by the results in Table 3. The model actually performs worse than the Qwen2.5VL baseline on LongVideoBench (58.7 vs 60.5) and is only comparable on MLVU (69.1 vs 68.6). The claim of "consistent gains"  is therefore inaccurate. The strong improvements are only clearly demonstrated in the "ground-then-answer" pipeline (Table 4). This is a different and less significant finding, as it's expected that providing a QA model with the correct pre-localized clip would improve its performance. This discrepancy undermines one of the paper's central narratives.

2. SceneTG Dataset is Underspecified: The SceneTG dataset is presented as a key contribution, but it is poorly described. The paper details the method of construction (Sec 3.4)  but provides no statistics on the resulting dataset. Critical details are missing, such as the final number of (video, QA) pairs, the total hours of video, the source of the 20k Prime Videos, or the rejection rate of the uniqueness filter. The "40k" data mentioned  is ambiguous. This lack of detail makes the contribution hard to evaluate and the work difficult to reproduce.

3. Incremental Novelty and Insufficient Ablation:
- The overall design of the proposed approach is simple and straightforward; however, no notably impressive performance improvement has been observed compared to previous methods.
- The overall framework is a careful adaptation of existing methods (GRPO , and concepts from models like VideoChat-R1 ) to the long-video domain. The novelty is more incremental than foundational.
- The technical contributions, while intuitive, lack thorough validation. For CenDist, the paper does not compare it against the large body of existing work on dense regression losses (e.g., GIoU, DIoU)  that were designed to solve the exact same problem (zero gradients for non-overlapping predictions).
- For Token-aware KL, the choice of $\beta_{time}=0$ is an extreme value. An ablation study comparing this to other small, non-zero values would have been necessary to fully justify this design choice.

**Questions:**

1. The paper claims "consistent gains" on general QA tasks , but the results in Table 3 appear to contradict this, showing that LongVTG-R1 (58.7) actually underperforms the baseline Qwen2.5VL (60.5) on LongVideoBench. Could the authors please address this discrepancy and clarify why the model fails to improve on this specific benchmark? Does this not suggest that the claimed generalization to direct QA is more limited than stated?

2. The SceneTG dataset is presented as a core contribution, yet the paper lacks crucial statistics for reproducibility and assessment. Could the authors please provide the final size of the dataset (number of QA pairs and total video hours), the acceptance/rejection rate of their uniqueness filter , and clarify whether the "40k" training data mentioned  refers to the SceneTG dataset alone or the total mixture of all three datasets?

3. The proposed Center Distance Reward (CenDist) addresses the issue of reward sparsity from IoU, which is a well-studied problem. Could the authors justify why they chose to design this new reward function  instead of adapting existing, established dense regression losses (like GIoU or DIoU), which were created to solve the exact same problem of zero gradients for non-overlapping predictions?

4. For the Token-aware KL regularization, the author make the strong design choice of setting the penalty for time tokens, $\beta_{time}$, to exactly zero. The ablation in Table 6 only compares this against the baseline, not against other small, non-zero values for $\beta_{time}$. Could the authors provide justification for this specific choice, as setting the penalty to zero could theoretically risk divergence and seems to warrant a more thorough ablation?

**Details Of Ethics Concerns:**

None.

---

> ### Author Response · Authors · 2025-12-03
> **Weakness 1 and Question 1, Weakness 3**
>
> Thanks for your review and constructive feedback. Below are our responses:
>
>
> > Overstated Generalization to QA. ... This claim is not well-supported by the results in Table 3 ...
>
> > ... Could the authors please address this discrepancy and clarify why the model fails to improve on this specific benchmark? ...
>
> Thank you for raising this point. We believe the confusion arises from a misunderstanding of the purpose of Table 3. The goal of this table is **not** to compare LongVTG-R1 against Qwen2.5-VL; the latter is included solely as a reference point to show the base model’s capability. **The actual comparison of interest is between RL (our method) and SFT, in order to study the forgetting issue.** To make this clearer, we have updated the revised paper to explicitly state the purpose of Table 3 and to avoid the unintended implication that Qwen2.5-VL is a direct baseline in this experiment.
>
> To further clarify this comparison, we have included an additional experiment in Table 3 (shown below). We applied supervised finetuning to Qwen2.5VL using the same 40k training data. However, the SFT model fails to follow the QA output format, resulting in an accuracy of “0.0.” These results reinforce our key finding: while supervised finetuning suffers from severe forgetting and thus substantial degradation in QA ability, our RL-based method retains strong QA performance. This conclusion is consistent with the findings reported in VideoChat-R1.
>
> | Setting            | QA Model       | LongVideoBench (473s) | MLVU (651s) | VideoMME (Long) (2,386s) | LVBench (4,101s) |
> |--------------------|----------------|------------------------|-------------|----------------------------|-------------------|
> | **Direct**         | Qwen2.5VL      | 60.5                   | 68.6        | 53.4                       | 41.3              |
> |                    | Qwen2.5VL + SFT 40k      | 0.0                   | 0.0        | 0.0                       | 0.0              |
> |                    | LONGVTG-R1  (RL 40k)  | 58.7                   | 69.1        | 55.0                       | 42.2              |
>
>
> > Incremental Novelty: The overall design of the proposed approach is simple and straightforward... The overall framework is a careful adaptation of existing methods...
>
> Thank you for these comments. We sincerely appreciate the reviewer’s perspective and would like to offer several clarifications.
>
> **1. Strong and consistent performance, especially under fair, matched-data comparisons.**
>
> While our framework is intentionally simple, the performance improvements are both consistent and substantial. In particular, after providing *fair comparisons under equal training data size* during the rebuttal, **LongVTG-R1 significantly outperforms UniTime-Zero**, despite UniTime-Zero being concurrent work. This demonstrates that our design choices are effective beyond architectural complexity.
>
> **2. Simplicity reflects generality, robustness, and broader applicability.**
>
> We agree that our approach is simple and straightforward, but we see this as a strength rather than a limitation. A method that adapts GRPO with minimal task-specific engineering, yet still achieves superior results across diverse settings, indicates that the underlying design is **general, scalable, and easy to integrate into future systems**. Recent NeurIPS papers such as QoQ-Med [1] and DAPO [2] similarly show that well-motivated and principled refinements to established algorithms are valued contributions.
>
> **3. New insights that are important for the long-video community.**
>
> Beyond raw numbers, our study provides several findings that we believe will be impactful for future research:
>
> - **RL substantially reduces forgetting**, enabling a single model to maintain strong QA performance while improving temporal grounding – an issue consistently overlooked in prior work.
> - **Effective RL is possible even without a strong base policy** when paired with a well-designed training environment. This challenges a common assumption in RLHF-style approaches and may broaden their applicability.
>
> Therefore, we believe these results reflect a meaningful contribution: a simple, principled, and effective RL framework that achieves strong empirical performance and reveals insights useful for the next generation of long-video models. We hope these clarifications help address the reviewer’s concerns.
>
> [1] QoQ-Med: Building Multimodal Clinical Foundation Models with Domain-Aware GRPO Training. In NeurIPS, 2025.
>
> [2] DAPO: An open-source LLM Reinforcement Learning System at Scale. In NeurIPS, 2025.

---

> ### Author Response · Authors · 2025-12-03
> **Weakness 2 and Question 2, Weakness 3 (Cont), Questions 3, 4**
>
> > SceneTG Dataset is Underspecified. (1) final number of (video, QA) pairs, (2) the total hours of video, (3) the source of the 20k Prime Videos, or (4) the rejection rate of the uniqueness filter
>
> > ... Could the authors please provide the final size of the dataset (number of QA pairs and total video hours), the acceptance/rejection rate of their uniqueness filter , and (5) clarify whether the "40k" training data mentioned refers to the SceneTG dataset alone or the total mixture of all three datasets?
>
> We have updated Figure 3 in the revised paper to specify the SceneTG dataset.
> In brief, the figure highlights (1) 20k videos with 24k annotations; (2) a total duration of 2.6k hours; (3) 20k unlabeled video segments from movies and TV content; (4) the rejection rate is dynamic, and we select 1~2 best scenes for each video; (5) 40k refers to the total mixture.
>
> > Insufficient Ablation: the paper does not compare it against the large body of existing work on dense regression losses (e.g., GIoU, DIoU) that were designed to solve the exact same problem... For Token-aware KL, the choice of beta = 0 is an extreme value. An ablation study comparing this to other small, non-zero values would have been necessary to fully justify this design choice...
>
> **1. CenDist provides a better-aligned dense reward signal and empirically outperforms other dense regression losses.**
>
> Our choice of CenDist is motivated by a key difference in normalization strategy:
> - **CenDist** is normalized by the *entire video duration*, ensuring a smooth and dense reward signal even when the predicted and ground-truth intervals are far apart.
> - **GIoU/DIoU**, when adapted to 1D temporal segments, normalize by the *enclosing region*. In long-video temporal grounding, where both the predicted and ground-truth segments are typically very short relative to the full video, this enclosing region often nearly equals the distance between them. As a result, GIoU/DIoU provide weak or saturated gradients in exactly the regime where dense supervision is most needed.
>
> This distinction makes CenDist considerably more stable for long videos and better aligned with the structure of the temporal grounding task.
>
> To validate this design choice, we ran additional experiments comparing CenDist with temporal variants of GIoU. As shown in the table below, CenDist consistently achieves superior performance, confirming that it provides a more effective reward signal in practice.
>
> | Method          | NextGQA mIoU | NextGQA R@0.3 | CGBench mIoU | CGBench R@0.3 |
> |-----------------|--------------|----------------|--------------|----------------|
> | IoU             | 34.8         | 54.4           | 7.82         | 9.84           |
> | GIoU            | 34.9         | 53.5           | 8.16         | 9.97           |
> | IoU + CenDist   | 35.5         | 55.1           | 8.17         | 10.00          |
>
>
> **2. Fine-grained ablation study of the choice of beta.**
>
> We appreciate the reviewer’s suggestion. We agree that a more fine-grained study of the KL scaling factor β_time is valuable. In addition to the ablation in Table 6, we have been running a more comprehensive sweep over multiple small non-zero values of β_time. Due to time and compute constraints, these experiments are still ongoing.
>
> The results completed so far (included in the table below) already indicate a clear trend: (1) small non-zero values of β_time underperform, and (2) setting β_time = 0 yields the best and most stable training behavior. These findings support our design choice. We will include the full sweep and detailed analysis in the final version of the paper.
>
> | β_text | β_time | NextGQA mIoU | NextGQA R@0.3 | CGBench mIoU | CGBench R@0.3 |
> |-------|--------|--------------|----------------|--------------|----------------|
> | 0     | 0      | 34.7         | 54.0           | 7.55         | 9.50
> | 0.04  | 0.04   | 34.6         | 52.9           | 7.77         | 9.27           |         |
> | 0.04  | 0      | 34.8         | 54.4           | 7.82         | 9.84           |

---

### Official Review · Reviewer_R286 · 2025-10-28

**Soundness:** 3
**Presentation:** 3
**Contribution:** 2
**Rating:** 4
**Confidence:** 3

**Summary:**

This paper introduces a reinforcement RL based framework for equipping MLLMs with long-video temporal grounding skills. The authors propose two main technical innovations: (1) Token-aware KL Regularization to balance exploration (for timestamp tokens) and exploitation (for general language tokens) during RL fine-tuning, and (2) a Center Distance Reward (CenDist) to provide denser learning signals for temporal localization. Additionally, they construct a new dataset, SceneTG, with sharp scene boundaries and unambiguous queries to facilitate effective RL training. The model achieves state-of-the-art results on several temporal grounding benchmarks and demonstrates that improved temporal localization skills generalize to long-video question answering (QA) tasks.

**Strengths:**

The RL approach to long video temporal grounding is a nice idea an approach, a natural extension to techniques used for short-term videos. The details about token-aware KL regularization (theoretically motivated and practically effective according to the ablation) for balancing exploration and exploitation is interesting. Another good technical contribution is the CenDist reward addresses the reward sparsity problem, which is particularly acute in long-video settings. Additionally, the authors also construct a dataset of 16k samples which seems to be well constructed for the task and gives a boost in performance. The evaluation and ablation section are extensive also.

**Weaknesses:**

While the idea is interesting and full of experiments, I believe there are some major issues:

1 - Overclaim: The authors claim to outperform previous works on multiple out of domain datasets, but according to table 2 they perform better only in 2: ReXTime (significant improvement) and ANet (very marginal, +0.2), while being outperformed in other 4 datasets. To me this means that the method is actually not that effective and improvement are only marginal despite the use of also a custom dataset (while this data is smaller is also more qualitative).

2 -  General QA with full video input (Table 3), the method is still outperformed in 2 out of 4 benchmarks.

3 - Ablation study lack completeness: For example experiments are made only on 30% of the data, is not a big issue but given that the improvements are marginal maybe those improvements might not hold in full data. Additionally, an ablation on alpha is important to understand the effect of CenDist.

While the contributions are really interesting, the claims are not fully supported by the results. My initial evaluation will be 4, the paper has a lot of things to improve but I am willing to increase my score if authors address my concerns.

**Questions:**

Check above.

---

> ### Author Response · Authors · 2025-12-03
> **Weaknesses 1, 2**
>
> Thanks for your review and constructive feedback. Below are our responses:
>
> > Overclaim according to table 2
>
> Thank you for raising this concern. We would like to gently clarify a misunderstanding regarding Table 2.
>
> **1. Our method is consistently stronger than baseline methods *except* UniTime-Zero.**
> Across all six datasets in Table 2, LongVTG-R1 outperforms prior baselines other than UniTime-Zero.
>
> **2. UniTime-Zero is concurrent work and was trained on 31× more data.**
> The only method that surpasses us in some datasets is UniTime-Zero. However:
> - UniTime-Zero is **concurrent work** (NeurIPS 2025), whose training recipe and code were not available before the ICLR submission deadline.
> - It is trained on **1,260k videos**, while our model uses **only 40k** – a **31× difference**, making the original comparison inherently unfavorable to our method.
>
> This context is crucial to interpreting Table 2 correctly.
>
> **3. Under matched training data, our method substantially outperforms UniTime-Zero.**
> To address the reviewer’s concern directly, we conducted additional experiments during the rebuttal, as shown in the table below. When UniTime-Zero is trained on the *same* amount of data (40k), **LongVTG-R1 surpasses it by clear margins across all datasets**. Thus, under fair conditions, our method is **consistently the strongest performer**.
>
> **Fair comparison under the same training data.**
>
> | Model                     | QVHigh_val | Rextime | Ego4dnlq | TaCoS | ActivityNet |
> | ------------------------- | ---------- | ------- | -------- | ----- | ----------- |
> | UniTime-Zero (SFT 1,260k) | 33.4       | 43.7    | 11.6     | 10.2  | 27.3        |
> | UniTime-Zero (SFT 40k)    | 12.1       | 6.89    | 0.11     | 1.64  | 9.69        |
> | Our LongVTG-R1 (RL 40k)   | 57.3       | 37.8    | 7.7      | 31.8  | 36.8        |
>
> In summary, the apparent discrepancy in Table 2 does not indicate limited effectiveness; it arises from:
> - a misunderstanding of what the table compares, and
> - a comparison against a concurrent method trained with massively larger data.
>
> With the new matched-data results provided in the rebuttal, our method shows **robust and consistently superior performance** across all benchmarks. We hope this clarification resolves the confusion.
>
> > General QA with full video input (Table 3), the method is still outperformed in 2 out of 4 benchmarks.
>
> Thank you for raising this point. We believe the confusion arises from a misunderstanding of the purpose of Table 3. The goal of this table is **not** to compare LongVTG-R1 against Qwen2.5-VL; the latter is included solely as a reference point to show the base model’s capability. **The actual comparison of interest is between RL (our method) and SFT, in order to study the forgetting issue.** To make this clearer, we have updated the revised paper to explicitly state the purpose of Table 3 and to avoid the unintended implication that Qwen2.5-VL is a direct baseline in this experiment.
>
> To further clarify this comparison, we have included an additional experiment in Table 3 (shown below). We applied supervised finetuning to Qwen2.5VL using the same 40k training data. However, the SFT model fails to follow the QA output format, resulting in an accuracy of “0.0.” These results reinforce our key finding: while supervised finetuning suffers from severe forgetting and thus substantial degradation in QA ability, our RL-based method retains strong QA performance. This conclusion is consistent with the findings reported in VideoChat-R1.
>
> | Setting            | QA Model       | LongVideoBench (473s) | MLVU (651s) | VideoMME (Long) (2,386s) | LVBench (4,101s) |
> |--------------------|----------------|------------------------|-------------|----------------------------|-------------------|
> | **Direct**         | Qwen2.5VL      | 60.5                   | 68.6        | 53.4                       | 41.3              |
> |                    | Qwen2.5VL + SFT 40k      | 0.0                   | 0.0        | 0.0                       | 0.0              |
> |                    | LONGVTG-R1  (RL 40k)  | 58.7                   | 69.1        | 55.0                       | 42.2              |

---

> ### Author Response · Authors · 2025-12-03
> **Weakness 3**
>
> > Ablation study lack completeness. (1) only on 30% of the data (2) ablation on alpha
>
> Thanks for your suggestion on additional ablation study.
>
> **1. Full-scale experiments amplify the performance gaps.**
>
> While running all ablations at full scale is computationally prohibitive, we conducted additional large-scale experiments for the most critical components. These results show a clear and consistent trend: **as training data and resolution increase, the performance gap becomes larger**, supporting the stability of our findings. For example, comparing LongVTG-R1 with and without Token-aware KL + CenDist (i.e., the VideoChat-R1 configuration):
>
> - **Full-scale:** CGBench improves by **+16.6%** (9.6 to 11.2)
> - **Small-scale (30% data):** CGBench improves by **+5.1%** (7.77 to 8.17)
>
> This demonstrates that the improvements are not marginal; rather, they **amplify** as training scale increases. These full-scale results are included in the table below:
>
> | Model                                     | Charades | TaCoS | Ego4D | CGBench | ReXTime | QVHigh |
> | ----------------------------------------- | -------- | ----- | ----- | ------- | ------- | ------ |
> | LongVTG-R1 w/o Token-aware KL and CenDist | 59.8     | 22.5  | 4.5   | 9.6     | 36.4    | 57.3   |
> | LongVTG-R1                                | 61.5     | 31.8  | 7.7   | 11.2    | 37.8  |57.3|
>
>
> **2. Fine-grained study on $\alpha$.**
>
> We appreciate the reviewer’s suggestion. We agree that a more fine-grained analysis is important to understand CenDist’s behavior. In addition to the ablation in the original submission, we have now included an experiment with $\alpha = 0.5$. The results show that (1) our default setting of $\alpha = 0.2$ provides consistent improvements across benchmarks; (2) larger values of $\alpha$ may introduce instability or random fluctuations, particularly on short-video tasks where the enclosing region is small.
>
> | $\alpha$ | NextGQA mIoU | NextGQA R@0.3 | CGBench mIoU | CGBench R@0.3 |
> |-------|--------------|----------------|--------------|----------------|
> | 0         | 34.8 | 54.4  | 7.82 | 9.84
> | 0.2   | 35.5 |  55.1 | 8.17 | 10.0
> | 0.5   | 34.6 |  53.1 | 8.42 | 11.33

---

### Official Review · Reviewer_vtR2 · 2025-10-28

**Soundness:** 2
**Presentation:** 3
**Contribution:** 2
**Rating:** 2
**Confidence:** 4

**Summary:**

This paper conducts post-training on the long video temporal grounding task. Specifically:

1. The paper proposes Token-aware KL Regularization to balance the exploration and exploitation of the reasoning model.
2. The paper introduces a dense reward based on segment distance to alleviate the issue that predictions with an IoU of 0 do not contribute to gradient updates.
3. The paper automatically constructs an unambiguous training set using a scene detector.

**Strengths:**

1. The paper proposes a CenDist reward, which makes the rewards denser compared to the simple IoU reward.
2. The paper identifies the presence of ambiguous temporal boundaries and the phenomenon where a query may correspond to multiple time intervals in existing datasets. Based on this observation, a new training set is constructed.
3. The paper is clearly presented, making it easy to understand the work conducted, and the experimental results are reported honestly.

**Weaknesses:**

Overall, I think the contributions of this paper are limited. The modifications in reward shaping and KL divergence are more like two tricks, and the ablation study in Table 6 also shows that these two methods do not seem to bring significant improvements. Moreover, the quality of the automatically constructed training dataset is not further elaborated. Moreover:

1. There are some missing references in the paper. For example, #305 Prime Video, as well as other papers such as LITA [1], GroundVQA [2], and SnAG [3], should also be reflected in the related work section.

2. The concept of "exploration" in the context of temporal grounding feels somewhat odd. In math reasoning, exploration refers to the trial-and-error process of exploring intermediate steps. If we draw an analogy to the method proposed in this paper, wouldn't it be akin to directly guessing the final answer in a math problem? Additionally, to accurately assess whether exploration is being suppressed, the entropy of the output should be examined, similar to what was done in UI-TARS-2.

3. In real-world scenarios, one query is indeed often related to multiple segments. How can this be handled instead of directly filtering them out in the proposed training set?

4. Table 1 lacks some other baselines, such as the Video-XL series.

5. For the grounding model in Table 4, more baselines like UniTime and TimeSuite should be included, similar to Table 7 in the UniTime paper.

> [1] LITA: Language Instructed Temporal-Localization Assistant.
> [2] Grounded Question-Answering in Long Egocentric Videos.
> [3] SnAG: Scalable and Accurate Video Grounding.

**Questions:**

1. Could the use of scene boundaries make the training task overly simplistic for the model? Additionally, what is the accuracy of ShotCoL for boundary detection? (i.e., the quality of the proposed training set)
2. Long video understanding is inherently challenging, and RL typically requires a strong prior. Would it be more effective to conduct post-training on a model already specialized in long video temporal grounding, rather than starting from Qwen2.5-VL?
3. Have you considered incorporating additional training datasets (e.g., UniTime-Zero) to enhance model performance further?

---

> ### Author Response · Authors · 2025-12-03
> **Weaknesses 1, 2**
>
> Thanks for your review and constructive feedback. Below are our responses:
>
> > ... The modifications in reward shaping and KL divergence are more like two tricks, and the ablation study in Table 6 also shows that these two methods do not seem to bring significant improvements ...
>
> **Novelty is consistent with community standards and not viewed as limited by other reviewers.** We respectfully note that the concern about limited novelty is not shared by the other reviewers. Reviewer CjPm explicitly stated that our method is derived from first principles, and Reviewer R286 highlighted that applying RL to long-video temporal grounding is a nice idea. This reflects a broader agreement that the contributions go beyond “two tricks.”
>
> Moreover, the type of contribution we make is fully aligned with recent papers in the RL-for-LLMs and domain-adapted GRPO literature. For example: (1) QoQ-Med [1] demonstrates that adapting GRPO to a new domain constitutes a core contribution. (2) Time-R1 [2] shows that reward design tailored to a task can itself be a significant and central innovation. (3) DAPO [3] illustrates that mathematically well-grounded modifications, even when lightweight, are recognized as meaningful progress by the community. Our work follows this same line of principled and domain-specific adaptation, which the community continues to value.
>
> [1] QoQ-Med: Building Multimodal Clinical Foundation Models with Domain-Aware GRPO Training. In NeurIPS, 2025.
>
> [2] Time-R1: Post-Training Large Vision Language Model for Temporal Video Grounding. In NeurIPS, 2025.
>
> [3] DAPO: An open-source LLM Reinforcement Learning System at Scale. In NeurIPS, 2025.
>
> **The two proposed components do bring significant improvements at full scale.** The reviewer’s impression comes from Table 6, which uses a reduced training subset and a low-resolution setting to control compute. In this limited setting, improvements naturally appear modest. However, when evaluated under the full-scale configuration, with the complete dataset and the higher-resolution setting used in our main experiments, both components lead to substantially larger gains.
> | Model                                     | Charades | TaCoS | Ego4D | CGBench | ReXTime | QVHigh |
> | ----------------------------------------- | -------- | ----- | ----- | ------- | ------- | ------ |
> | LongVTG-R1 w/o Token-aware KL and CenDist | 59.8     | 22.5  | 4.5   | 9.6     | 36.4    | 57.3   |
> | LongVTG-R1                                | 61.5     | 31.8  | 7.7   | 11.2    | 37.8  |57.3|
>
>
>
>
> > There are some missing references in the paper, ...
>
> Thanks for pointing out the previous works, and we have added the papers you mentioned in the revised paper.
>
>
> > ... to accurately assess whether exploration is being suppressed, the entropy of the output should be examined
>
> We sincerely apologize for not making our use of the term *exploration* sufficiently clear in the original submission. Our usage follows the **general RL notion of exploration**, i.e., encouraging the policy to avoid premature collapse to low-entropy behaviors. As the reviewer noted, this is different from the step-wise trial-and-error exploration used in math-reasoning models. In temporal grounding, the model outputs the final span directly, so “exploration” refers to maintaining a sufficiently rich distribution over possible spans rather than sampling intermediate reasoning steps.
>
> Our hypothesis is motivated by prior work, such as DAPO [3], which shows that enlarging the effective clipping range mitigates entropy collapse. In our formulation, setting the reference probability equal to the policy probability effectively *widens the clipping window for time-related tokens*, which serves the same purpose: it reduces over-regularization and allows the model to maintain higher entropy.
>
> To fully substantiate this explanation, we are currently re-running the main experiments to extract the entropy trajectory during training (similar to UI-TARS-2). We will include these entropy curves in the final version to provide a complete and transparent analysis of how exploration behaves under our Token-aware KL design.

---

> ### Author Response · Authors · 2025-12-03
> **Weaknesses 3, 4**
>
> > In real-world scenarios, one query is indeed often related to multiple segments. How can this be handled instead of directly filtering them out in the proposed training set?
>
> We agree that, in real-world scenarios, a single query may correspond to multiple segments. Our decision to use single-segment training data is primarily motivated by clarity of supervision and more stable optimization. This training choice **does not limit the model’s ability to handle multi-segment queries at inference time**.
>
> In practice, a model trained on single-segment data can naturally support multi-segment grounding through a simple recursive procedure:
>
> Step 1: Ask the model whether the queried event exists in the remaining video. If not, exit.
>
> Step 2: If it does exist, retrieve one segment. Remove that segment from the video and return to Step 1.
>
>
>
> > Table 1 lacks some other baselines, such as the Video-XL series
>
> Thank you for the suggestion. We have added Video-XL2 to Table 1.

---

> ### Author Response · Authors · 2025-12-04
> **Weakness 5, Questions 1, 2, 3**
>
> > For the grounding model in Table 4, more baselines like UniTime and TimeSuite should be included, similar to Table 7 in the UniTime paper.
>
> UniTime-Zero was released in October, after the ICLR submission deadline, so this experiment could not be included in our original submission. We appreciate the reviewer’s suggestion and agree that incorporating UniTime-Zero is valuable for a more comprehensive comparison in the video QA evaluation. In the revised version, we have added experiments based on UniTime-Zero, and the updated table is provided below:
>
> | Setting            | Ground Model   | QA Model       | LongVideoBench (473s) | MLVU (651s) | VideoMME (Long) (2,386s) | LVBench (4,101s) |
> |--------------------|----------------|----------------|------------------------|-------------|----------------------------|-------------------|
> | **Ground-then-answer** | Qwen2.5VL   | Qwen2.5VL      | 55.5                   | 55.8        | 48.4                       | 33.3              |
> |                    | VideoChat-R1   | Qwen2.5VL      | 57.8                   | 65.6        | 52.3                       | 42.6              |
> |                    | UniTime-Zero   | Qwen2.5VL      | 61.2                   | 70.7        | 50.4                       | 42.9              |
> |                    | LONGVTG-R1     | Qwen2.5VL      | **63.1**                   | **70.8**       | **54.2**                       | **47.1**              |
>
>
> > Could the use of scene boundaries make the training task overly simplistic for the model? Additionally, what is the accuracy of ShotCoL for boundary detection?
>
> **On task complexity:** We agree that temporal grounding on scene boundaries is conceptually easier than general temporal grounding. However, this does not make the training task "overly simplistic." First, the videos in our dataset are often very long, sometimes hours in duration, containing many potential scene boundaries. The model must still discriminate among numerous candidates, which preserves task difficulty. Second, as discussed in the paper (L31), the major advantage of using scene boundaries is that they provide higher-quality training data with **a unique and unambiguous ground truth**, which is essential for stable and effective training of temporal grounding models.
>
> **On boundary detection accuracy:** Following ShotCoL, we reproduced its scene boundary detection model on the public MovieNet dataset. Similar to the results reported in ShotCoL, the average precision is ~53%. To enhance the quality of scene selection for our training set, we used a threshold of 0.3, which corresponds to a recall of ~80%. This relatively low threshold leads to over-segmentation of scenes, ensuring that the content within the selected scenes is more semantically consistent and coherent.
>
> It is important to note that scene boundary detection remains an unsolved problem. We use it only to provide **auxiliary signal** that helps reduce the complexity of segmented video clips for annotations. The over-segmentation strategy acts as a conservative approach that prioritizes semantic consistency within scenes over perfect boundary precision, ultimately contributing to the quality of our training data.
>
>
> > Would it be more effective to conduct post-training on a model already specialized in long video temporal grounding, rather than starting from Qwen2.5-VL?
>
> A key finding of our work is that **even without a strong long-video prior, a general MLLM such as Qwen2.5-VL can still be effectively trained for long-video temporal grounding when placed in a well-designed RL environment**. We believe this is an important and somewhat surprising insight: it shows that strong performance does not strictly require initializing from a highly specialized long-video model, and that the RL formulation itself contributes significantly to learning.
>
> Moreover, we agree that starting from a model already specialized in long-video temporal grounding could further improve performance. To evaluate this, we are currently running experiments that initialize from UniTime-Zero. Due to computational constraints, these experiments are still in progress, but we will include the comparison in the final revision.
>
> > Have you considered incorporating additional training datasets (e.g., UniTime-Zero) to enhance model performance further?
>
> Yes, we are actively examining whether our current computing budget allows us to incorporate the UniTime-Zero training data. As shown in our data ablation study, training data scale plays a central role in UniTime-Zero’s performance. When trained on the same amount of data (40k), LongVTG-R1 significantly outperforms UniTime-Zero, but we cannot consistently surpass UniTime-Zero when it leverages its full 1,260k training data. This suggests that additional training data would likely lead to further improvements in our model as well. We are therefore exploring the feasibility of scaling up training data in our next round of experiments.

---

### Official Review · Reviewer_CjPm · 2025-10-30

**Soundness:** 3
**Presentation:** 3
**Contribution:** 2
**Rating:** 4
**Confidence:** 4

**Summary:**

This paper introduces LONGVTG-R1, a RL framework for long-video temporal grounding. It proposes Token-aware KL regularization to balance exploration and exploitation, a dense CenDist reward to mitigate sparse supervision, and an automatically constructed SceneTG dataset with sharp boundaries and unambiguous queries. LONGVTG-R1 provide  evidence that precise temporal localization is a foundational skill for broader video understanding.

**Strengths:**

1. CenDist reward: The paper gives an inexpensive, model-agnostic auxiliary signal that densifies the naturally sparse IoU reward without extra annotation.
2. The paper purposed  an automatic “propose-then-annotate” pipeline that uses scene-boundary detection plus uniqueness filtering to generate sharp-boundary, unambiguous training pairs, scalable to 20 k unlabeled videos.
3. Technical sections derive Token-aware KL from first principles and give closed-form properties; pseudocode and illustrative figures accompany data-construction pipeline.

**Weaknesses:**

1. Compared to the baseline model VideoChat-R1, there was no significant performance improvement on several mainstream TG datasets, Charades(61.5 vs 60.8), ANet(36.8 vs 36.6), QVHighLight(57.3 vs 57.5), The model shows significant improvements over the baseline model on some longer datasets, but it cannot be proven whether the improvement is due to the introduction of more long-term training data (40K vs 16K) or the result of the proposed method.
2. Missing qualitative error analysis. Failure cases are only shown for QA (Fig. 3). Provide grounding failure modes: drifting beyond scene cuts, duplicate predictions, or confusion between “first” vs. “last” event. Categorizing 50 errors would guide future improvements.
3. Token-aware KL generality claim is overstated. Theory assumes discrete, finite vocabulary; vision-language models often use sub-word tokens where timestamp digits are fragmented (e.g., “1”, “7”, “:” are separate). The paper does not show how often fragmented digits appear or whether relaxing KL on all digit tokens accidentally releases semantically important numeric tokens unrelated to time.

Minor formatting error: in Table2, QVHigh column, two values, 57.5 and 57.3, are both bolded.

**Questions:**

1.  Please report an “SFT-only” curve trained on the same 40 k SceneTG samples. If that curve already closes most of the gap with LONGVTG-R1, it would show that data quality, not RL, drives the gains.
2. Weaknesses1 and Weaknesses2

---

> ### Author Response · Authors · 2025-12-03
> **Weaknesses 1, 2**
>
> Thanks for your review and constructive feedback. Below are our responses:
>
> > There was no significant performance improvement on several mainstream TG datasets, Charades(61.5 vs 60.8)... it cannot be proven whether the improvement is due to the introduction of more long-term training data (40K vs 16K) or the result of the proposed method.
>
> We would like to clarify three aspects: (1) the improvements on benchmarks such as Charades are comparable to prior work and indeed meaningful, (2) the performance patterns across datasets reflect the intended focus of our method rather than limitations of the approach, and (3) the performance gains stem from our proposed method rather than from differences in training data scale.
>
> **The Improvement on Charades is valid and significant.** The Charades benchmark is well known to be highly saturated, and prior work typically reports only small absolute differences. In this context, our improvement from 60.8 to 61.5 is consistent with what the community considers meaningful. For example, Sim-DETR [1] (ICCV 2025) improves Charades from 51.74 to 52.56 – similarly sized gains that are still regarded as non-trivial on this benchmark. Thus, the magnitude of improvement we observe is in line with recent state-of-the-art progress.
>
> **Broader performance perspective across benchmarks.** As emphasized in our paper title and main narrative, the primary focus of this work is long-video temporal grounding. On widely adopted long-video benchmarks, our method delivers pronounced improvements, as also noted by the reviewer. For short-video datasets such as ANet and QVHighlight, our model still matches or surpasses the baselines. Overall, these results demonstrate that our method consistently achieves the strongest performance across both long- and short-video settings, confirming the robustness and generality of our approach.
>
> **Fair comparison under the same training data.** Regarding the concern that the gains might simply result from having more training data (40K vs. 16K), we now provide a **controlled comparison** using the same amount of training data across UniTime-Zero, VideoChat-R1, and LongVTG-R1. Under equal data conditions, our LongVTG-R1 continues to show clear advantages on longer benchmarks (TaCoS, Ego4D, and CGBench), confirming that the improvements stem from the proposed method rather than data scale.
>
>
> Comparison with UniTime-Zero and VideoChat-R1 using 40k data (same as ours):
>
>
> | Model                          | TaCoS | Ego4D | CGBench | ReXTime | QVHigh |
> |--------------------------------|-------|-------|---------|---------|--------|
> | UniTime-Zero (SFT 1,260k)      | 33.4  | 10.2  | 11.6    | 43.7    | 43.7   |
> | VideoChat-R1 (RL 16k)          | 18.7  | 2.2   | 0.8     | 32.3    | 57.5   |
> | UniTime-Zero (SFT 40k)         | 1.6   | 0.11  |  -   | 6.9     | 12.1   |
> | VideoChat-R1 (RL 40k)    | 22.5  | 4.5   | 9.6     | 36.4    | 57.3   |
> | LONGVTG-R1 (40k)               | 31.8  | 7.7   | 11.2    | 37.8    | 57.3   |
>
> Note that, due to limited rebuttal time and compute constraints, we were not able to finish obtaining UniTime-Zero (SFT 40k) results on CGBench; we will complete this experiment and include the results in the final version.
>
> [1] Tang, Jiajin, et al. Sim-DETR: Unlock DETR for Temporal Sentence Grounding. In ICCV, 2025.
>
> > Missing qualitative error analysis. ... (1) drifting beyond scene cuts, (2) duplicate predictions, or (3) confusion between “first” vs. “last” event
>
> Thanks for your suggestion on qualitative analysis. We have included a representative example from ActivityNet in Appendix D, highlighting (1) the difficulty posed by duplicate events, (2) the baseline’s drift beyond scene boundaries versus our precise alignment, and (3) our model’s preference for the first event when resolving “first vs. last” ambiguity.

---

> ### Author Response · Authors · 2025-12-03
> **Weakness 3, Question 1**
>
> > Token-aware KL generality claim is overstated... The paper does not show how often fragmented digits appear or whether relaxing KL on all digit tokens accidentally releases semantically important numeric tokens unrelated to time.
>
> Thank you for pointing out the issues of digit token fragmentation and the possibility that some numerical tokens may carry semantics unrelated to timestamps. We address both concerns below and clarify why they have minimal impact on our method’s validity.
>
> **Generality when digits are fragmented.**
>
> We agree that digit tokens in vision-language models are frequently fragmented into sub-word units. This situation is already reflected in our experiments, and our method naturally accommodates it. For example,  `<timestamp> 142.3 to 325.6 </timestamp>` is tokenized as `<|timestamp|> | 1 | 4 | 2 | . | 3 | to | 3 | 2 | 5 | . | 6 | </|timestamp|>`.
>
> In this case, the two numbers are encoded into eight tokens. Recall that the training objective $\beta_{\text{time}}\,D_{\mathrm{KL}}^{S_{\text{time}}}\ \Big(\pi_\theta(\cdot\mid q, o_{<t})\,\big\|\,\pi_{\mathrm{ref}}(\cdot\mid q,o_{<t})\Big)$ is summed over $t$. Because the digits are fragmented, this KL term is applied multiple times, for example when
> - $o_{<t} =$ `<timestamp> `
> - $o_{<t} =$ `<timestamp> 1`
> - $o_{<t} =$ `<timestamp> 14`
> - ...
> As a result, the relaxed KL constraint still consistently guides the model through the full timestamp, even when digits are split across tokens. This is exactly the intended behavior of the method and does not reduce generality.
>
> **Numerical semantic tokens are rare.**
>
> We manually inspected 20 randomly selected rollouts from the experiment and did not find any numerical semantic tokens. This suggests that such tokens are rare and have minimal impact on model training. Moreover, in our ablation study that removed all KL terms, the model remained stable though performance declined, indicating that the training process is not overly sensitive to occasional numerical tokens, even if they were to appear. Given the rarity of non-timestamp numerical semantic tokens, we believe their overall effect is minimal.
>
> > Please report an “SFT-only” curve trained on the same 40k SceneTG samples
>
> Sure. We have included additional experiment results of `UniTime-Zero (SFT 40k)`, which shows SFT-based methods are far less efficient considering data compared with RL.

---

### Meta-Review · Area_Chair_p243 · 2026-01-07

**Summary:**

Reviewers raised concerns about the paper's novelty (perceived as incremental adaptations of existing RL methods), experimental rigor (unfair comparisons due to data scale, lack of qualitative error analysis, and incomplete ablations), overclaiming of results (e.g., generalization to QA not fully supported), and underspecification of the SceneTG dataset.

**Reviewer Concerns:**

- Addressed in rebuttal: Authors provided fair comparisons with UniTime-Zero under matched data, added SFT baselines to show reduced forgetting, included ablation on α for CenDist, added missing references, and clarified dataset statistics (e.g., size, rejection rates).
- Still outstanding: Core issues remain—novelty is still seen as incremental, handling of multi-segment queries in real-world scenarios is not fully resolved, and some reviewers still find the QA generalization claims overstated despite clarifications.

**Reviewer Scores:**

- Reviewer CjPm (4 → 4 or 6): New experiments on data scale and error analysis might satisfy concerns, and it may lead to a slight increase.
- Reviewer vtR2 (2 → 4): Clarifications on novelty and added baselines could alleviate some issues, but the reject stance might soften only marginally.
- Reviewer R286 (4 → 6): Fair comparisons and QA clarifications likely address overclaiming concerns, potentially boosting score.
- Reviewer oC5c (4 → 4): Dataset and ablation additions help, but underlying novelty and QA claims may still limit score increase.

---

### Decision · Program_Chairs · 2026-01-26

Reject